# Multidisciplinary Management of Suspected Lyme Borreliosis: Clinical Features of 569 Patients, and Factors Associated with Recovery at 3 and 12 Months, a Prospective Cohort Study

**DOI:** 10.3390/microorganisms10030607

**Published:** 2022-03-12

**Authors:** Alice Raffetin, Julien Schemoul, Amal Chahour, Steve Nguala, Pauline Caraux-Paz, Giulia Paoletti, Anna Belkacem, Fernanda Medina, Catherine Fabre, Sébastien Gallien, Nicolas Vignier, Yoann Madec

**Affiliations:** 1Department of Infectious Diseases, Tick-Borne Diseases Reference Center-Paris and Northern Region, General Hospital Lucie et Raymond Aubrac, 94190 Villeneuve-Saint-Georges, France; amal.chahour@chiv.fr (A.C.); steve.nguala@chiv.fr (S.N.); pauline.caraux-paz@chiv.fr (P.C.-P.); anna.belkacem@chiv.fr (A.B.); fernanda.medina@chiv.fr (F.M.); sebastien.gallien@aphp.fr (S.G.); 2European Study Group for Lyme Borreliosis (ESGBOR), ESCMID, Gerbergasse 14 3rd Floor, 4001 Basel, Switzerland; 3EA 7380 Dynamyc, Université Paris-Est Créteil, Ecole Nationale Vétérinaire d’Alfort, USC Anses, 94000 Créteil, France; 4Groupe de Recherche et d’Etude des Maladies Infectieuses-Paris Sud-Est (GREMLIN Paris Sud-Est), 94000 Créteil, France; nicolas.vignier@ghsif.fr; 5Laboratoire de Santé Animale USC EPIMAI, Anses, Ecole Nationale Vétérinaire d’Alfort, 94700 Maisons-Alfort, France; 6Department of Rheumatology, Tick-Borne Diseases Reference Center-Paris and Northern Region, General Hospital Lucie et Raymond Aubrac, 94190 Villeneuve-Saint-Georges, France; julien.schemoul@chiv.fr; 7Department of Public Health, Groupe Hospitalier Sud Ile-de-France, 77000 Melun, France; 8Department of Psychiatry, Tick-Borne Diseases Reference Center-Paris and Northern Region, General Hospital Lucie et Raymond Aubrac, 94190 Villeneuve-Saint-Georges, France; giulia.paoletti@chiv.fr; 9Department of Neurology, Tick-Borne Diseases Reference Center-Paris and Northern Region, General Hospital Lucie et Raymond Aubrac, 94190 Villeneuve-Saint-Georges, France; catherine.fabre@chiv.fr; 10Department of Infectious Diseases and Clinical Immunology, University Hospital Henri Mondor, 94000 Créteil, France; 11Centre d’Investigation Clinique Antilles Guyane, CIC Inserm 1424, Centre Hospitalier de Cayenne, 97300 Cayenne, France; 12Department of Social Epidemiology, Institut Pierre Louis d’Épidémiologie et de Santé Publique, IPLESP, Inserm UMR 1136, Sorbonne Université, 75012 Paris, France; 13Department of Infectious Diseases, Hôpitaux Universitaires Paris Seine-Saint-Denis, CHU Avicenne, APHP, Université Sorbonne Paris Nord, 93000 Bobigny, France; 14Epidemiology of Emerging Diseases Unit, Institut Pasteur, 75015 Paris, France; yoann.madec@pasteur.fr

**Keywords:** lyme borreliosis, multidisciplinary management, serial misdiagnosis

## Abstract

**Introduction**. Because patients with a suspicion of Lyme borreliosis (LB) may have experienced difficult care paths, the Tick-Borne Diseases Reference Center (TBD-RC) was started in 2017. The aim of our study was to compare the clinical features of patients according to their final diagnoses, and to determine the factors associated with recovery in the context of multidisciplinary management for suspected LB. **Methods**. We included all adult patients who were seen at the TBD-RC (2017–2020). Four groups were defined: (i) confirmed LB, (ii) possible LB, (iii) Post-Treatment Lyme Disease Syndrome (PTLDS) or sequelae, and (iv) other diagnoses. Their clinical evolution at 3, 6, and 9–12 months after care was compared. Factors associated with recovery at 3 and at 9–12 months were identified using logistic regression models. **Results**. Among the 569 patients who consulted, 72 (12.6%) had confirmed LB, 43 (7.6%) possible LB, 58 (10.2%) PTLDS/sequelae, and 396 (69.2%) another diagnosis. A favorable evolution was observed in 389/569 (68.4%) at three months and in 459/569 (80.7%) at 12 months, independent of the final diagnosis. A longer delay between the first symptoms and the first consultation at the TBD-RC (*p* = 0.001), the multiplicity of the diagnoses (*p* = 0.004), and the inappropriate prescription of long-term antibiotic therapy (*p* = 0.023) were negatively associated with recovery, reflecting serial misdiagnoses. **Conclusions**. A multidisciplinary team dedicated to suspicion of LB may achieve a more precise diagnosis and better patient-centered medical support in the adapted clinical sector with a shorter delay, enabling clinical improvement and avoiding inappropriate antimicrobial prescription.

## 1. Introduction

Lyme borreliosis (LB) is the most common tick-borne disease in Europe and the USA, caused by spirochetes of the *Borrelia burgdorferi* sensu lato complex [1,2]. In 2018, the annual incidence in France was 104 cases/100,000 inhabitants (95% Confidence Interval (CI) 91–117), demonstrating a constant increase since 2014 [3,4].

Clinical diagnosis of LB may be difficult because of its wide range of clinical pictures, sometimes resembling other pathologies (rheumatological diseases, auto-immune diseases, neurological disorders etc.) The most frequent clinical manifestations in Europe are erythema migrans (EM) and Lyme neuroborreliosis (LNB) [5]. Some functional symptoms may be present at all stages, which can further complicate the diagnosis [6,7]. Such symptoms may persist after a well-conducted treatment following the guidelines [post-treatment Lyme disease syndrome (PTLDS)] [5,6,8]. Rare sequelae causing definitive impairment may occur [5,6,9]. Rare coinfections with LB, transmitted by a tick-bite, are also described [10,11].

Microbiological diagnosis of LB relies on a two-tier serological test and PCR, for which sensitivities and specificities depend on the disease stage and the anatomical site sampled [12,13,14,15,16]. Current diagnostic tools are performant if their indication and interpretation are well-respected; otherwise, they may lead to an incorrect diagnosis.

Treatment of LB relies on antibiotic therapy for 10–28 days according to its stage and its clinical manifestation [15,17]. No studies have yet proven to be of benefit for longer treatment [18,19,20,21,22]. Nonetheless, there are no clear guidelines for the management of functional, persistent symptoms, which sometimes leaves patients unrelieved.

Therefore, because patients with a suspicion of LB may have experienced diagnostic delay and difficult care paths [23,24,25,26], we started a multidisciplinary LB center at the end of 2017, which is a joint endeavor of the departments of infectious diseases, internal medicine, rheumatology, neurology, algology, dermatology, psychiatry, microbiology, and physical rehabilitation. Our center was named the Tick-borne Diseases Reference Center (TBD-RC) for Paris and Northern Region in July 2019 by the French Ministry of Health. Other teams in other countries have also created such care organizations [27,28,29,30], showing a European awareness for the management of complex LB and its differential diagnoses. Several studies have been published to describe these new care organizations, but none have compared the clinical features of patients according to their diagnosis nor have they described patient care paths and outcomes after multidisciplinary and patient-centered medical support.

The aim of our study was to compare the clinical features of patients attending the TBD-RC according to their diagnosis (LB or not), to describe their care paths and outcomes, and to determine the factors associated with recovery in the context of multidisciplinary management for suspected LB.

## 2. Materials and Methods

We conducted a prospective descriptive and analytical cohort study, including all adult patients who consulted the TBD-RC for a suspicion of LB, from 1 December 2017 to 1 December 2020.

### 2.1. Population, Setting, and Intervention

For management at the TBD-RC, a medical file and a letter from a physician who referred the patient was requested prior to consultation, enabling the team to analyze all previous consultations, hospitalizations, and performed tests. Data on previous treatments were collected. Non-recommended treatments were defined as an antibiotic therapy longer than eight weeks and/or associated antimicrobials (≥2 prescribed concomitantly). There were no limitative criteria to receive patients, especially regarding positive or negative *Borrelia* serology. After a dedicated and multidisciplinary one-hour consultation with a meticulous physical exam, a medical summary was made and a first orientation offered in a one-day hospitalization, a conventional hospitalization, or an outpatient management. If indicated, a serological test for *Borrelia* was prescribed as well as a cerebrospinal fluid analysis for LNB or articular analysis for Lyme arthritis [16,31]. Other tests (a search for other tick-borne diseases, autoimmune disorders, etc.) were performed if clinically relevant. A complementary expert medical evaluation was requested if needed.

Patients with LB-associated symptoms were classified as follows [5,30,32,33]: (i) confirmed LB (tick-exposure, typical clinical signs, and a positive two-tiered serology), (ii) possible LB (tick exposure and/or prior EM, evocative clinical signs, and marked clinical improvement after 21 days of antibiotics), (iii) PTLDS (asthenia/polyalgia/cognitive complaints) or sequelae (objective and definitive impairment after a LNB, an ACA or a Lyme arthritis) persisting for more than six months after proven LB had been treated as recommended [15,33]. Patients not fulfilling these definitions were considered in the group “other diagnoses”. All diagnoses were made by a physician who specialized in the corresponding field. All the complex cases were discussed during a multidisciplinary consultation meeting to refine the diagnosis.

Finally, patient-centered care in the adapted medical department was offered to all patients to treat any disease/symptoms—even without a definitive diagnosis. Antibiotic therapy was prescribed if the patient presented an untreated confirmed or possible LB according to the guidelines [15,33]. Management was re-evaluated through a medical consultation at 3, 6, and 9–12 months to confirm its accuracy and adapt it if necessary. Patients who were living far away and could have adapted care from their general practitioner (GP) or a specialized physician were only re-evaluated at 9–12 months at the TBD-RC (consultation or teleconsultation). At each evaluation, a clinical statement was made by the doctor according to the patient’s point of view (joint oral conclusion): complete recovery, partial improvement (persistent clinical signs or symptoms allowing resumption of daily and professional activities), stagnation, or deterioration.

### 2.2. Patient Data

We collected patient data in standardized medical files (tick-exposure, past history of tick-bite, past-history of erythema migrans, delay between the tick-exposure and the symptoms, detailed clinical signs and symptoms, serological results for LB, past history of treatments etc.) at the TBD-RC independently of the study. Anxiety and sadness were measured with the MADRS scale, STAI form, and QIDS-SR16 scale, and asthenia with the FSS-11 score [34,35,36].

### 2.3. Statistical Analysis

The four groups of patients were compared according to socio-demographic, clinical and microbiological characteristics, and 3- and 9–12-month outcomes after multidisciplinary care.

Categorical variables are reported as proportions and percentages, and continuous variables as median with interquartile range (IQR). Categorical variables were compared by chi-squared or Fischer’s exact test as appropriate. Continuous variables were compared between groups by ANOVA as appropriate.

Factors associated with rapid recovery (evaluated at three months) and with recovery at a later point in time (evaluated at 9–12 months) were identified using logistic regression models. In both analyses, factors associated with the outcome with a *p*-value < 0.25 in univariate analysis were considered in the multivariate model. A stepwise backward regression was used to identify factors that remained independently associated with the outcome. Gender, age, and “group of patient” were forced in the models.

A *p*-value *<* 0.05 was defined for statistical significance. All analyses were performed using Stata version 16 (College Station, TX, USA).

### 2.4. Approval of the Ethics Committee

The local ethics committee of the University Intercommunal Hospital of Créteil, France, gave its approval for this research. All included patients gave their consent to use their medical data for research purposes prior to their management at the TBD-RC. The research sponsor signed a commitment to comply with the “Reference Methodology MR004” of the French Data Protection Authority, CNIL declaration number 2216096v0 (10 December 2019).

## 3. Results

### 3.1. Comparison of the Clinical and Epidemiological Characteristics of the Patients

During the study period, 569 patients consulted the TBD-RC of Paris and Northern region for suspicions of LB. Based on clinical and biological criteria and a multidisciplinary evaluation, 72/569 (12.7%) fulfilled criteria for confirmed LB, 43/569 (7.6%) possible LB, 58/569 (10.2%) PTLDS/sequelae, and 396/569 (69.2%) another diagnosis (Table 1 and Table 2, among whom 51 (9.0%) had no specific diagnosis but confirmed LB, possible LB, and PTLDS/sequelae were ruled out. Moreover, other diagnoses associated with confirmed LB (but unrelated with LB) were found in 36/72 (50%), with possible LB in 30/43 (69.8%), and with PTLDS/sequelae in 34/58 (58.6%). These differential and associated diagnoses are described Table 2. Among the 569 patients, 298 (52.4%) had a single diagnosis, 159 (27.9%) had two diagnoses, and 99 (17.4%) had more than three diagnoses retained. A mean of 1.7 diagnoses/patient was made with a median delay of 15.5 [IQ25,75 = 0;82] days at the TBD-RC.

The comparative epidemiological characteristics of the patients are presented in Table 3. The median (IQR) delay between onset of symptoms and first consultation at the TBD-RC was 1.4 (0.4–3.8) years and was significantly shorter for confirmed LB (0.3 years or 4 months) (*p* < 0.001). Only 180/569 (31.6%) patients had a two-tiered positive serological test for *Borrelia.* Patients with a negative test and confirmed or possible LB had symptoms for less than six weeks. Prior to consulting the TBD-RC, 5/569 (0.9%) had a positive test only in Western-Blot, and all of these had another diagnosis. Prior to care at the TBD-RC, at least one antibiotic therapy had been prescribed in 369/569 (64.9%), and a non-recommended antibiotic therapy in 101/569 (17.8%). The median (IQR) duration of previous antibiotic therapy was three (two to seven) weeks. The proportion of patients who underwent non-recommended antibiotic therapy was significantly larger in the PTLDS/sequelae group (*p* = 0.001). The main clinical signs are presented in Table 4.

Care paths for the 569 patients are presented Figure 1. All patients were offered patient-centered care in the adapted medical department, mainly in the departments of infectious diseases, rheumatology, and psychology. Antibiotic therapy was prescribed at the TBD-RC for 148/569 (26%) patients for a median (IQR) duration of three (three to four) weeks. A total of 504/569 patients had a planned follow-up at the TBD-RC at three, six, and 9–12 months, and 65/569 had just one reevaluation at 9–12 months (Figure 1 and Figure 2).

### 3.2. Factors Associated with Rapid Recovery

At three months, 484/504 (96.0%) patients were evaluated. The proportion of patients with complete recovery, partial improvement, stagnation, or deterioration differed among the four diagnostic groups (*p* = 0.001). The proportions of patients with rapid recovery were 29/72 (40.3%), 9/43 (20.9%), 8/58 (13.8%), and 84/396 (21.2%) in confirmed LB, possible LB, PTLDS/sequelae, and other diagnoses, respectively (Figure 2).

Factors associated with rapid recovery in univariate and multivariate analysis are presented in Table 5 Factors independently associated with lower odds of rapid recovery were longer delay between onset of symptoms and the first consultation at the TBD-RC (*p* = 0.001), longer delay to final diagnosis (*p* < 0.001), multiplicity of diagnoses (*p* = 0.004), and a history of non-recommended antibiotic therapy (*p* = 0.023). A history of antibiotics use was not associated with recovery (*p* = 0.50 in univariate analysis). A first line of antibiotics prescribed at the TBD-RC was associated with recovery (*p* = 0.036 in univariate analysis). The odds of rapid recovery did not differ among study groups.

In a sensitivity analysis at three months, patients with rapid recovery were compared to those with partial improvement only. Results were similar to the ones described above (Appendix A).

### 3.3. Factors Associated with Recovery at a Later Point in Time

At 9–12 months, 528/569 (92.8%) patients were evaluated. Twenty-eight patients were lost to follow-up, and 13 had an unknown status as they had not yet been re-evaluated at 9–12 months. In the absence of clinical evaluation at 9–12 months, patients with a complete recovery at three and/or six months were considered in recovery at 9–12 months. The proportions of patients with complete recovery, partial improvement, stagnation, or deterioration differed among the four groups (*p* = 0.001). The proportions of patients with recovery at a later point in time were 51/72 (70.8%), 20/43 (46.5%), 19/58 (32.8%), and 125/396 (31.6%) in confirmed LB, possible LB, PTLDS/sequelae, and other diagnoses, respectively (Figure 2).

Factors associated with a recovery at a later point in time in univariate and multivariate analysis are presented in Table 6. Factors independently associated with lower odds of recovery were longer delay between onset of symptoms and the first consultation at the TBD-RC (*p* < 0.001), and a history of non-recommended antibiotic therapy (*p* = 0.05). The odds of recovery were significantly higher for those identified as having confirmed LB (*p* = 0.004). The odds of recovery were significantly higher for those identified as having confirmed LB (*p* = 0.004) and in those with a history of EM (*p* = 0.020). After adjusting for other factors, multiplicity of diagnoses was no longer associated with lower odds of recovery, neither was antibiotic treatment provided at the TBD-RC with higher odds of recovery. A history of antibiotics use before consulting the TBD-RC was not associated with recovery (*p* = 0.52 in univariate analysis) as well as the prescription of a second line of antibiotics at the TBD-RC (*p* = 0.64 in univariate analysis).

### 3.4. Description of Patients with Stagnation or Deterioration in the Groups with a Primary Diagnosis of LB at 3 Months

In the group “confirmed LB”: two presented stagnation and one deterioration (compressive neurinoma, misuse of doxycycline, and rapid progression of a lymphoma). In the group “possible LB”: five presented stagnation (two previous neurodegenerative diseases that worsened, a rheumatoid arthritis, a Biermer’s disease, and a cirrhosis). In the group “PTLDS/sequelae”: three presented stagnation and one deterioration (three still had persistent neuropathic pains two years after confirmed LNB treated as recommended, with a negative repeated lumbar punction; and an acute arthrosis). These patients could not improve because of another co-existing disease than LB, diagnosed thanks to the initial multidisciplinary management.

## 4. Discussion

### 4.1. Summary of the Principal Findings

Among the 569 patients who consulted the TBD-RC for a suspicion of LB, 72 (12.6%) had confirmed LB, 43 (7.6%) possible LB, 58 (10.2%) PTLDS/sequelae, and 396 (69.2%) another diagnosis. Over the entire follow-up, favorable evolution was observed in most of the patients: 389/569 (68.4%) had completely recovered or partially improved allowing resumption of daily and professional activities at three months, and 459/569 (80.7%) at 9–12 months, independent of the diagnosis. Patients with partial improvement, stagnation, or deterioration presented associated diagnoses, explaining the absence of complete recovery. The main factors negatively associated with rapid recovery were longer delay between onset of symptoms and the first consultation at the TBD-RC, multiplicity of diagnoses, and inappropriate prescription of antibiotic therapy, reflecting serial misdiagnosis and the complexity of the cases. The diagnostic delay and the history of non-recommended antibiotic therapy as negative factors for recovery were confirmed at 9–12 months. A confirmed LB was associated with a better recovery at 9–12 months.

### 4.2. Similar Multidisciplinary Experiences in France and Europe

Patients seeking care at the TBD-RC presented similar epidemiological and clinical characteristics to those in other studies in other settings describing a multidisciplinary care organization for suspected LB [27,28,29,30]. As previously found in these studies, ~10–20% of suspected cases had confirmed LB.

The multiplicity of other diagnoses found in all these studies highlights the complexity of diagnosing LB without disregarding other diagnoses [27,28,29,30]. There is not one disease that presents one picture. This multiplicity was associated with lower odds of recovery at three months. A multidisciplinary care organization may achieve a more precise diagnosis and better patient-centered medical support, with a shorter delay as demonstrated here and in other studies [29], enabling clinical improvement.

Moreover, on one hand, there were not any statistical differences among the recovery of the four groups at three months, which suggests that a multidisciplinary approach is beneficial for all groups and not just one. On the other hand, confirmed LB was associated with better recovery at 9–12 months, suggesting that LB needs a long follow-up for better management of the patients and that patients with confirmed LB have more chances to be cured even if for some presentations it may take more than three months until improvement.

Few patients (9%) had no specific diagnoses compared to other studies varying from 12.1% to 38.5% [27,28,29,30], probably because they had a regular follow-up in the adapted care sector since the beginning of their management, which allowed for clinical re-evaluations and the confirmation of the diagnosis for a longer period. For patients that have experienced diagnostic delay or serial misdiagnosis, a multidisciplinary approach could be a major answer, helping with acceptance of the diagnosis and offered care [23].

### 4.3. Meaning of the Study and Implication for Practice

Functional symptoms were the most frequent in the four groups of patients (asthenia, polymyalgia, and polyarthralgia) with no significant difference, but objective symptoms significantly differentiated them (Table 4). The predominance of functional symptoms in patients with a suspicion of LB had been demonstrated in previous studies, as well as had facial palsy in confirmed LB [7,27,28,29,30], but no studies had compared the clinical signs of the four groups. Arthritis of small joints in PTLDS suggests an interesting connection between the inflammatory processes linked to previous LB (reversible) or chronic inflammatory rheumatism [37,38].

#### 4.3.1. Serology Does Not Rule the Diagnosis of LB

One third of the patients had a two-tiered positive serology. Among them, 74/180 (41.1%) had another diagnosis, and 34/180 (18.9%) PTLDS. Among patients with negative serology, 7/276 (2.5%) had proven LB, and 15/276 (5.4%) possible LB, and all had presented symptoms for less than six weeks. Our results are similar to the other descriptive cohort studies of multidisciplinary management [29,30,31,32]. Diagnosis of LB also relies on tick exposure and evocative clinical signs that are inseparable within the serology to confirm or refute the diagnosis [5,13,33].

#### 4.3.2. Non-Recommended Antibiotic Therapies Are Associated with a Poorer Clinical Evolution

If past antibiotic use was not associated with recovery, prolonged or associated antibiotic therapies were deleterious. The proportion of patients with non-recommended antibiotic therapy was significantly higher in the group PTLDS/sequelae, reflecting the difficulty of their management and suggesting a causative role of long-term antibiotics on persistent symptoms. Patients clinically improved when antibiotics were stopped and multidisciplinary care started. Five randomized trials have demonstrated the absence of benefit of prolonged antibiotic therapies [18,19,20,21,22].

A favorable evolution for the majority of the patients lets the physicians reassure them despite previous misdiagnosis or diagnostic delay. We specifically focused on complete recovery, which was more common in those with confirmed LB, probably because their disease was well identified and well taken care of. However, the majority of patients in all groups experienced partial recovery after seeking care at the TBD-RC, indicating the benefit of multidisciplinary care (Figure 2).

#### 4.3.3. A Longer Delay between the Onset of Symptoms and the First Consultation at the TBD-RC Is Associated with a Poorer Evolution

Early medical management enables better clinical outcome of the patients. Physicians should address their patients as soon as possible in these multidisciplinary structures, which seem to allow a better management and end diagnostic delay, taking time to listen to the patients, to examine, explore, and consult the expertise of different specialists. This raises the question whether or not these structures should be expanded for other diseases responsible for diagnostic delay, especially when we see here the low number of LB and TBD. General multidisciplinary clinics might also be an answer.

### 4.4. Strengths and Limitations of the Study

To our knowledge, this is the first study that compares clinical and epidemiological characteristics of patients who presented with suspected LB, based on the final diagnosis, to describe their clinical evolution within a multidisciplinary care center and to determine the factors associated with recovery. Moreover, we have presented one of the largest cohort studies of patients, and few were lost to follow-up, providing higher quality statistical analyses.

The first limitation is the study’s monocentric aspect, but it is qualified, as our results are similar to other multidisciplinary experiences in different European settings [27,29,30]. The second limitation is the inclusion of patients during the COVID-19 pandemic, which interrupted the flux of the patients between 15 March and 30 April 2020. We had no medical demand within this period, which reflected the dramatic decrease in the number of outpatient consultations for all medical fields. Normal activities resumed in May 2020. During the second lockdown (November 2020), 13 new patients with clinical emergencies (e.g., neurological signs) were evaluated, and we continued to follow up with all previous patients.

## 5. Conclusions

In our study, patients presented similar epidemiological and clinical characteristics to those in other studies in other settings describing a multidisciplinary care organization for suspected LB, but this is the first study which compared the clinical features and evolution of these four groups of patients (confirmed LB, possible LB, PTLDS/sequelae, and other diagnoses). Confirmed LB represented only a small percentage of patients (12.6%) who attended the TBD-RC of Paris and Northern Region. A favorable evolution was observed in the majority of the patients (80.7%) at 12 months, independent of the final diagnosis. A multidisciplinary care organization may achieve a more precise diagnosis and patient-centered medical support in the adapted clinical sector, with a shorter delay and avoiding inappropriate antibiotic therapies. For patients that experience serial misdiagnosis or diagnostic delay, a multidisciplinary approach could be a major answer, which would help with the acceptance of the diagnosis and the offered care.

## Figures and Tables

**Figure 1 microorganisms-10-00607-f001:**
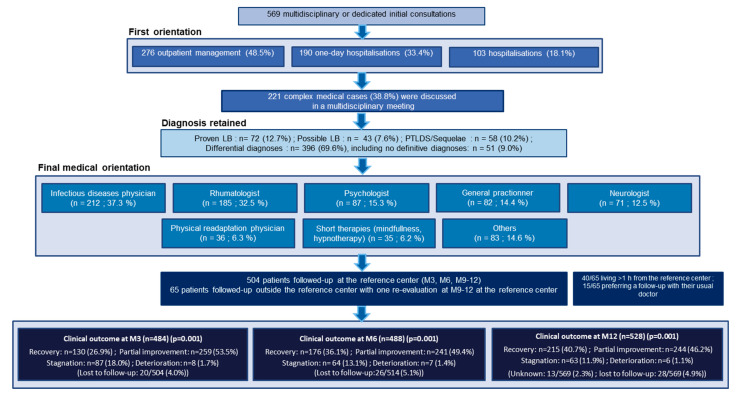
Care paths at the Tick-Borne Diseases Reference Center-Paris and Northern Region.

**Figure 2 microorganisms-10-00607-f002:**
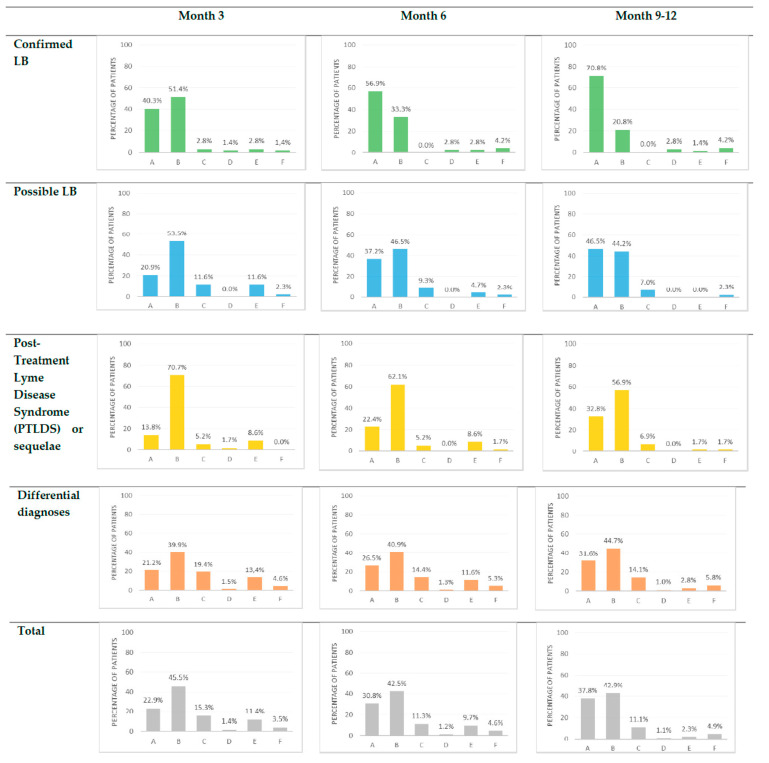
Clinical outcome of the patients consulting at TBD-RC of Paris and Northern region at three, six and 9–12 months. A = Recovery; B = Partial improvement; C = Stagnation; D = Deterioration; E = Unknown; F = Lost to follow up.

**Table 1 microorganisms-10-00607-t001:** Description of the final diagnoses made at the TBD-RC of Paris and Northern region.

Final Diagnoses			Diagnoses Implicating LB (N = 249/569)						Diagnoses with No Links with LB (N = 320/569)
	*Confirmed LB (n,%)*	*Possible LB* *(n,%)*		*PTLDS/Sequelae * (n,%)*		*Other Diagnoses (n,%)*			*Other Differential Diagnoses (n,%)*
				*PTLDS*	*Sequelae of LB*	*Failure of the Antibiotics Test ***	*Complete Recovery of a Treated LB*	*Monitoring after a Tick-Bite*	
**Total (N = 569, 100%)**	72 (12.7)	43 (7.6)		51 (9.0)	7 (1.2)	24 (4.2)	39 (6.9)	13 (2.3)	320 (56.2)
**EM**	26 (4.6) ***	-		14 (2.5) ****	-	-	-	-	-
**Lymphocytoma**	2 (0.4)	-		-	-	-	-	-	-
**Early LNB**	17 (3.0)	8 (1.4)		9 (1.6)	3 (0.5)	2 (0.4)	-	-	-
**Early Lyme arthritis**	2 (0.4)	3 (0.5)		-	1 (0.2)	-	-	-	-
**Early disseminated non-specific LB**	2 (0.4)	3 (0.5)		8 (1.4)	-	2 (0.4)	-	-	-
**Late LNB**	16 (2.8)	19 (3.3)		6 (1.1)	3 (0.5)	7 (1.2)	-	-	-
**Late Lyme arthritis**	0 (0)	2 (0.4)		-	-	1 (0.2)	-	-	-
**Early cardiac LB**	4 (0.7)	-		-	-	-	-	-	-
**ACA**	3 (0.5) *****	-		-	-	-	-	-	-
**Late disseminated non-specific LB**	-	6 (1.1)		4 (0.7)	-	3 (0.5)	-	-	-
**Unknown**	-	2 (0.4)		11 (1.9)	-	9 (1.6)	-	-	-

LB = Lyme borreliosis; EM = Erythema migrans; LNB = Lyme neuroborreliosis; ACA = Acrodermatitis chronica atrophicans; PTLDS = Post-Treatment Lyme Disease Syndrome. * Previous confirmed LB with sequelae or PTLDS; ** Initial suspicion of probable LB but patients did not improve after the antibiotic therapy, which led us to consider another diagnosis than LB; *** Among EM in confirmed LB: 6 isolated EM; 16 associated with non-specific symptoms such as asthenia, paresthesia etc., two associated with a LA or a LNB, 2 multiple EM; **** Among previous EM in PTLDS: 13 EM with initial non-specific symptoms and one multiple EM; ***** Among ACA in confirmed LB: one isolated, one associated with a peripheral neuropathy, one associated with cardiac conduction disturbances.

**Table 2 microorganisms-10-00607-t002:** Detailed description of the differential or associated diagnoses made at the TBD-RC of Paris and Northern region.

Other Infectious Diseases	68/569 (12.0)
Other tick-borne diseases (rickettsiosis, tularemia etc.)	9 (1.6)
Other bacterial infections (cutaneous infectious, tuberculosis, pneumonia etc.)	14 (2.5)
Viral infections (Epstein Barr Virus, Herpes Virus, Cytomegalovirus etc.)	22 (3.9)
Parasitic infections (larva migrans, schistosoma, toxocara etc.)	10 (1.8)
Post-infectious syndrome	13 (2.3)
**Rheumatological and auto-immune diseases**	**228/569 (40.1)**
Chronic inflammatory rheumatism (spondylarthritis, rheumatoid arthritis etc.)	55 (9.7)
Arthrosis and complications	59 (10.4)
Tunnel syndrome	47 (8.3)
Tendinopathy	24 (4.2)
Other rheumatological diseases	12 (2.1)
Auto-immune diseases (Gougerot-Sjogren disease, multiple sclerosis, lupus etc.)	31 (5.5)
**Neurological disorders**	**109/569 (19.2)**
Peripheral neuropathy	26 (4.6)
Dementia	10 (1.8)
Optical neuritis	5 (0.9)
Sequelae of stroke	5 (0.8)
Others (parkinsonism, Charcot’s disease etc.)	10 (1.8)
**Vitamin deficiencies (B9, D, PP, C etc.)**	**98/569 (17.2)**
**Psychiatric disorders**	**68/569 (12.0)**
Anxiety and/or depression	43 (7.6)
Psychotic disorders	11 (1.9)
Panic disorder	6 (1.1)
Others (addiction, post-traumatic syndrome, bipolar disorders etc.)	14 (2.5)
**Iatrogenism linked to a prolonged antibiotic therapy**	**65/569 (11.4)**
**Bodily Distress Syndrome**	**52/569 (9.1)**
**Endocrinopathy (thyroid disorders, adrenal disorders etc.)**	**21/569 (3.7)**
**Others (cancers, sleep apnea syndrome, genetic diseases, cardiovascular diseases etc.)**	**67/569 (11.8)**
**No specific diagnosis**	**51/569 (9.0)**

**Table 3 microorganisms-10-00607-t003:** Comparison of the epidemiological characteristics of the patients consulting the TBD-RC of Paris and Northern region.

Epidemiological Characteristics of the Patients	Total N = 569 (%)	Confirmed LB N = 72 (%)	Possible LB N = 43 (%)	PTLDS or Sequelae N = 58 (%)	Other Diagnoses N = 396 (%)	*p*-Value
**Age, years (median [IQ 25,75])**	48 (35.61)	52.5 (36.65)	52 (46.59)	47.5 (36.64)	47 (34.60)	0.14
**Male**	220 (38.7)	42 (58.3)	19 (44.2)	15 (25.9)	144 (36.4)	0.001
**Life style**						0.74
Home in a rural area	121 (21.2)	12 (16.7)	13 (30.2)	14 (24.1)	82 (20.7)	
Employment in rural areas/forest	30 (5.3)	4 (5.6)	1 (2.3)	3 (5.2)	22 (5.6)	
Forest-based leisure activities	399 (70)	55 (76.4)	28 (65.1)	40 (69.0)	276 (69.7)	
No exposure	20 (3.5)	1 (1.4)	1 (2.3)	1 (1.7)	16 (4.0)	
**Past history of tick-bite**	372 (65.3)	59 (81.9)	33 (76.7)	46 (79.3)	234 (59.1)	<0.001
**Past history of erythema migrans**	145 (25.4)	39 (54.2)	18 (41.9)	25 (43.9)	64 (16.2)	<0.001
**Patients referred by a physician** **with a letter**	516 (90.7)	69 (95.8)	42 (97.7)	51 (87.9)	354 (89.4)	0.016
General Practitioner	401 (70.4)	46 (63.9)	36 (83.7)	46 (79.3)	273 (68.9)	
Specialist physician	94 (16.5)	17 (23.6)	4 (9.3)	5 (8.6)	68 (17.2)	
Emergency unit physician	21 (3.7)	6 (8.3)	2 (4.7)	0 (0.0)	13 (3.3)	
No letter, patient self-referral	53 (9.5)	3 (4.2)	1 (2.33)	7 (12.1)	42 (10.6)	
**Duration (days) of chief complaints prior to examination at TBD-RC** **(median [IQ 25,75])**	512 [156,1392.5]	123.5 (37,233)	296 (132,1138)	374.5 (167,1078)	735 (219,1778)	<0.001
**Patient’s chief complaint**						<0.001
**Erythema migrans**	17 (3)	8 (11.1)	0 (0.0)	1 (1.7)	8 (2.0)	
Clinical signs/symptoms implicating early disseminated LB (>six months)	159 (27.9)	40 (55.6)	17 (39.5)	19 (32.8)	83 (21.0)	
Clinical signs/symptoms implicating late disseminated LB (>six months)	382 (67.2)	24 (33.3)	26 (60.5)	38 (65.5)	294 (74.2)	
Questions after a tick-bite	6 (1.1)	0 (0.0)	0 (0.0)	0 (0.0)	6 (1.5)	
Positive serological test with no clinical signs	5 (0.9)	0 (0.0)	0 (0.0)	0 (0.0)	5 (1.26)	
**Serological test**						<0.001
IgM and/or IgG positive in ELISA and WB	180 (31.6)	54 (75.0)	18 (41.9)	34 (58.6)	74 (18.7)	
IgG positive in ELISA only	75 (13.2)	5 (6.9)	10 (23.3)	9 (15.5)	51 (12.9)	
IgM and IgG negative in ELISA	276 (48.5)	7 (9.7)	15 (34.9)	15 (25.9)	239 (60.4)	
No serology (suspicion of erythema migrans)	38 (6.7)	6 (8.3)	0 (0.00)	0 (0.00)	32 (8.1)	
**Antibiotic therapy prescribed before TBD-RC**	369 (64.9)	51 (70.8)	27 (62.8)	58 (100.0)	233 (58.8)	<0.001
Antibiotic therapy > four weeks	117 (22.6)	15 (20.8)	4 (9.3)	29 (50.0)	69 (17.4)	<0.001
Non-recommended treatments (>eight weeks of antibiotics and/or associated antimicrobials)	101 (17.8)	7 (9.7)	1 (2.3)	23 (39.7)	70 (17.7)	<0.001

LB = Lyme borreliosis; PTLDS = Post-Treatment Lyme Disease Syndrome; ELISA = Enzyme-Linked Immunosorbent Assay; WB = Western-Blot; TBD-RC = Tick-Borne Diseases Reference Center.

**Table 4 microorganisms-10-00607-t004:** Comparison of the main clinical signs and symptoms presented by the patients consulting at TBD-RC of Paris and Northern region at baseline in the four different groups.

Clinical Signs	Total N = 569 (%)	Confirmed LB N = 72 (%)	Possible LB N = 43 (%)	PTLDS or Sequelae N = 58 (%)	Other Diagnoses N = 396 (%)	*p*-Value
**Polymyalgia**	213 (37.4)	22 (30.6)	17 (39.5)	21 (36.2)	153 (38.6)	0.611
**Polyarthralgia**	300 (52.7)	34 (47.2)	22 (51.2)	30 (51.7)	214 (54.0)	0.749
**Asthenia**	380 (66.8)	48 (66.7)	35 (81.4)	48 (82.8)	249 (62.9)	0.004
**Fever, chills**	49 (8.6)	2 (2.8)	7 (16.3)	1 (1.7)	39 (9.9)	0.014
**Night Sweat**	50 (8.8)	1 (1.4)	3 (7.0)	3 (5.2)	43 (10.9)	0.043
**Paresthesia**	225 (39.5)	27 (37.5)	26 (60.5)	21 (36.2)	151 (38.1)	0.035
**Headache**	141 (35.6)	26 (36.1)	18 (41.9)	24 (41.4)	209 (36.7)	0.740
**Insomnia**	96 (16.9)	11 (15.3)	7 (16.3)	9 (15.5)	69 (17.4)	0.959
**Loss of weight**	72 (12.7)	3 (4.2)	7 (16.3)	4 (6.9)	58 (14.7)	0.039
**Arthritis-small joints**	39 (6.9)	2 (2.8)	0 (0.0)	7 (12.1)	30 (7.6)	0.050
**Arthritis-large joints**	71 (12.5)	8 (11.1)	9 (20.9)	9 (15.5)	45 (11.4)	0.275
**Facial palsy**	19 (3.4)	10 (13.9)	1 (2.3)	4 (6.9)	4 (1.0)	0.001
**Neuropathic pain**	130 (22.9)	22 (30.6)	18 (41.9)	14 (24.1)	76 (19.2)	0.003
**Memory impairment**	99 (17.4)	8 (11.1)	11 (25.6)	13 (22.4)	67 (16.9)	0.167
**Concentration impairment**	94 (16.5)	5 (6.9)	11 (25.6)	14 (24.1)	64 (16.2)	0.020
**Radicular pain**	62 (10.9)	7 (9.7)	10 (23.3)	8 (13.8)	37 (9.3)	0.039
**Spinal pain**	116 (20.4)	11 (15.3)	10 (23.3)	7 (12.1)	88 (22.2)	0.198
**Vertigo**	85 (14.9)	5 (6.9)	9 (20.9)	9 (15.5)	62 (15.7)	0.171
**Anxiety**	103 (18.1)	9 (12.5)	11 (25.6)	4 (6.9)	79 (20.0)	0.030
**Sadness**	66 (11.6)	5 (6.9)	10 (23.3)	6 (10.3)	45 (11.4)	0.062
**Psychotic disorders**	24 (4.2)	0 (0.0)	2 (4.7)	0 (0.0)	22 (5.6)	0.058
**Cardiac conduction disturbances**	5 (0.9)	3 (4.2)	0 (0.0)	0 (0.0)	2 (0.5)	0.015

LB = Lyme borreliosis; PTLDS = Post-Treatment Lyme Disease Syndrome.

**Table 5 microorganisms-10-00607-t005:** Univariate and multivariate analyses of the associated factors with recovery versus partial improvement or stagnation or deterioration at three months after care at the TBD-RC-Paris and Northern region.

Risk Factor	N (*n* = 484)	*n*(%) Rapid Recovery at 3 Months	Crude OR [95% CI]	*p*-Value	Adjusted OR [95% CI]	*p*-Value
**Age (years)**				0.23		0.26
<35	127	39 (30.7)	1		1	
35–48	121	25 (20.7)	0.59 (0.33–1.05)		0.56 (0.29–1.06)	
48–61	114	29 (25.4)	0.77 (0.44–1.36)		0.72 (0.38–1.38]	
>61	122	37 (30.3)	0.98 (0.57–1.68)		0.95 (0.51–1.76]	
**Sex**				0.17		0.98
Male	188	57 (30.3)	1.33 (0.88–2.00)		1.01 (0.63–1.60]	
Female	296	73 (24.6)	1		1	
**History of tick-bite**				0.19	-	-
Yes	328	94 (28.7)	1.34 (0.86–2.08)			
No	156	36 (23.1)	1			
**History of erythema migrans**				0.023	-	-
Yes	120	43 (35.8)	1.67 (1.08–2.58)			
No	331	76 (23.0)	1			
**Serology**				0.004	-	-
Positive serology in ELISA and WB	154	45 (29.2)	1.51 (0.95–2.40)			
Positive serology in ELISA only	65	19 (29.2)	1.51 (0.81–2.79)			
Negative serology in ELISA	237	51 (21.5)	1			
Patient with no serology (erythema migrans)	28	15 (53.6)	4.21 (1.88–9.41)			
**Delay 1st symptoms-1st consultation at the TBD-RC**				<0.001		0.001
0–155 days (0.0–0.4 year)	131	53 (40.5)	1		1	
155–512 days (0.4–1.4 years)	129	41 (31.8)	0.69 (0.41–1.14)		0.86 (0.48–1.52)	
512–1393 days (1.4–3.8 years)	115	24 (20.9)	0.39 (0.22–0.69)		0.51 (0.26–0.97)	
>1393 days (>3.8 years)	108	12 (11.1)	0.18 (0.09–0.37)		0.22 (0.10–0.47)	
**Delay 1st consultation at the TBD-RC-final diagnosis**				<0.001		<0.001
0 day	189	80 (42.3)	1		1	
1–15 days	30	10 (33.3)	0.68 (0.30–1.53)		0.62 (0.26–1.49)	
15–83 days	142	23 (16.2)	0.26 (0.15–0.45)		0.27 (0.15–0.49)	
>83 days	123	17 (13.8)	0.22 (0.12–0.39)		0.28 (0.15–0.53)	
**Final diagnosis**				0.008		0.22
Confirmed LB	69	29 (42.0)	2.08 (1.21–3.56)		0.93 (0.49–1.77)	
Possible LB	37	9 (24.3)	0.92 (0.42–2.03)		1.13 (0.46–2.75)	
PTLDS or sequelae	53	8 (15.1)	0.51 (0.23–1.13)		0.40 (0.17–0.96)	
Other diagnoses	325	84 (25.9)	1		1	
**Number of diagnosis per patient**				<0.001		0.004
1 diagnosis	254	93 (36.6)	1		1	
2 diagnoses	139	24 (17.3)	0.36 (0.22–0.60)		0.43 (0.25–0.75)	
≥3 diagnoses	91	13 (14.3)	0.29 [0.15–0.55)		0.46 (0.23–0.93)	
**Antibiotics prescribed before the TBD-RC**				0.50	-	-
Yes	317	82 (25.9)	0.87 (0.57–1.32)			
No	167	48 (28.7)	1			
**History of non-recommended antibiotics**				<0.001		0.023
Yes	83	10 (12.1)	0.32 (0.16–0.64)		0.41 (0.19–0.88)	
No	401	120 (29.9)	1		1	
**First line of antibiotics prescribed at the TBD-RC**				0.036	-	-
Yes	140	47 (33.6)	1.59 (1.03–2.44)			
No	344	83 (24.1)	1			
**Second line of antibiotics at the TBD-RC**				0.75	-	-
Yes	17	4 (23.5)	0.83 (0.27–2.60)			
No	467	126 (27.0)	1			

TBD-RC = Tick-Borne Diseases Reference Center; PTLDS = Post-Treatment Lyme Disease Syndrome.

**Table 6 microorganisms-10-00607-t006:** Univariate and multivariate analyses of the associated factors with recovery versus partial improvement or stagnation or deterioration at 12 months after care at the TBD-RC-Paris and Northern region.

Risk Factor	N (*n* = 528)	*n*(%) Cured Patients at 12 Months	Crude OR [95% CI]	*p*-Value	Adjusted OR [95% CI]	*p*-Value
**Age (years)**				0.50		0.41
<35	138	60 (43.5)	1		1	
35–47	126	50 (39.7)	0.86 (0.52–1.40)		0.78 (0.46–1.34)	
48–61	129	46 (35.7)	0.72 (0.44–1.18)		0.65 (0.37–1.12)	
>61	135	59 (43.7)	1.01 (0.63–1.63)		0.93 (0.55–1.58)	
**Gender**				0.022		0.17
Male	200	94 (47.0)	1.52 (1.06–2.17)		1.31 (0.89–1.95)	
Female	328	121 (36.9)	1		1	
**History of tick-bite**				0.017	-	-
Yes	352	156 (44.3)	1.58 (1.08–2.30)			
No	176	59 (33.5)	1			
**History of EM**				<0.001		0.020
Yes	143	77 (53.9)	2.08 (1.41–3.07)		1.70 (1.09–2.65)	
No	384	138 (35.9)	1		1	
**Serology**				0.028	-	-
Positive serology in ELISA and WB	170	82 (48.2)	1.79 (1.20–2.66)			
Positive serology in ELISA only	70	30 (42.9)	1.44 (0.84–2.47)			
Negative serology in ELISA	254	87 (34.3)	1			
Patient with no serology (erythema migrans)	34	16 (47.1)	1.71 (0.83–3.51)			
**Delay 1st symptoms-1st consultation at the TBD-RC**				<0.001		<0.001
0–154 days (0.0–0.4 year)	131	78 (59.5)	1		1	
155–511 days (0.4–1.4 years)	138	67 (48.6)	0.64 (0.40–1.04)		0.82 (0.49–1.37)	
512–1393 days (1.4–3.8 years)	130	44 (33.9)	0.35 (0.21–0.58)		0.47 (0.28–0.81)	
>1393 days (>3.8 years)	128	26 (20.3)	0.17 (0.10–0.30)		0.26 (0.14–0.46)	
**Delay 1st consultation at the TBD-RC-final diagnosis**				0.064	-	-
0 day	228	107 (46.9)	1			
1–14 days	33	14 (42.4)	0.83 (0.40–1.74)			
15–83 days	133	45 (33.8)	0.58 (0.37–0.90)			
>83 days	134	49 (36.6)	0.65 (0.42–1.01)			
**Final diagnosis**				<0.001		0.004
Confirmed LB	68	51 (75.0)	5.69 (3.15–10.26)		3.13 (1.64–5.96)	
Possible LB	42	20 (47.6)	1.72 (0.91–3.28)		1.34 (0.67–2.68)	
PTLDS or sequelae	56	19 (33.9)	0.97 (0.54–1.76)		0.85 (0.44–1.62)	
Other diagnoses	362	125 (34.5)	1		1	
**Number of diagnosis per patient**				0.005	-	-
one diagnosis	286	134 (46.9)	1			
two diagnoses	149	53 (35.6)	0.63 (0.42–0.94)			
≥three diagnoses	93	28 (30.1)	0.49 (0.30–0.81)			
**Antibiotics prescribed before the TBD-RC**				0.52	-	-
Yes	345	137 (39.7)	0.89 (0.62–1.28)			
No	183	78 (42.6)	1			
**History of non-recommended antibiotics**				<0.001		0.05
Yes	96	25 (26.0)	0.45 (0.27–0.73)		0.58 (0.34–1.01)	
No	432	190 (44.0)	1		1	
**First line of antibiotics prescribed at the TBD-RC**				<0.001	-	-
Yes	143	84 (58.7)	2.76 [1.86–4.09)			
No	385	131 (34.0)	1			
**Second line of antibiotics at the TBD-RC**				0.64	-	-
Yes	15	7 (46.7)	1.28 (0.46–3.59)			
No	513	208 (40.6)	1			

TBD-RC = Tick-Borne Diseases Reference Center; PTLDS = Post-Treatment Lyme Disease Syndrome; EM = Erythema migrans.

## Data Availability

Data are available by sending an email at alice.raffetin@chiv.fr.

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
