# Peer review of "Multidisciplinary Management of Suspected Lyme Borreliosis: Clinical Features of 569 Patients, and Factors Associated with Recovery at 3 and 12 Months, a Prospective Cohort Study"

_microorganisms, 2022, doi:10.3390/microorganisms10030607_

Round 1

Reviewer 1 Report

This is a very interesting and well-designed study.

Comments, proposal for changes

Title

- Lyme borreliosis instead of Lyme borrelioses.

- The study reports on the outcome not only at 12 months but also at 3 months. Proposal: omit “at 12 months” or change to “at 3 and 12 months”.

Abstract

- Lines 35 and 36: Please change “Because many patients with a suspicious of Lyme borreliosis (LB) experience difficult care path …” to ~ “because many patients interpreted to have Lyme borreliosis…” or to “Because patients with a suspicion of Lyme borreliosis may have...”.

- Line 42 and elsewhere: Probably it would be better to use term other diagnoses instead of differential diagnosis.

Introduction

Line 72: Current recommendations for the duration of LB treatment are 10–28 days.

Methods

- Lines 110–115: While delineation into 3 groups is acceptable and seems logical, it is not in (complete) accord with the references 15 and 32.  

Please define PTLDS  (only symptoms are given but not their duration and impact on daily life) and sequelae.

- Lines 115 –117: Please re-write the sentence beginning with “Patients not fulfilling this definitions…”.

Results

- Line 161: … fulfilled criteria for confirmed LB instead of “presented confirmed LB”.

- Line 180, Table 1: Probably Table 1 and not Table 3, Table 1.

Table 1 is rather complicated and the current version may be misleading.

What does “Failure of antibiotic testing” mean?

Are Failure of antibiotic testing, Complete recovery of a treated LB and Monitoring after a tick bite a part of “Diferential diagnoses”?

Several abbreviations not used in the table are explained (EM, LNB, LA, ACA).

What does “conduction disorder” mean? Please specify.

- Line 188, Table 1 bis: Probably Parasitic infection instead of Parasitological infection; ?Bodily Distress Syndrome; ?parkinsonian syndrome???

- Line 191, Table 2: Probably No exposure instead of No exposition

Patient chief complain: It would be of interest to show also the duration of chief complaints prior to examination at TBD-RC.

Which clinical signs implicate early disseminated LB? Did the authors mean only signs or signs/symptoms?

Which clinical signs implicate late disseminated LB? Did the authors mean only signs or signs/symptoms?

Serological tests: No report on using two ELISAs?

Please explain all abbreviations (LB, PTLDS).

- Line 195; Table 3: In the column »Clinical signs« not only signs but also symptoms are depicted. What does »Radiculalgia« and »Rachialgia« mean?

What does »Conduction disorders« mean? Please specify: nerve conduction disorders, hearth conduction disorders, …?

 Please explain abbreviations.

- Line 197: Department of psychology or department of psychiatry?

- Line 201, Figure 1: Order of information shown in boxes is not logical.

- Line 208: “3.2. Factors associated to rapid recovery”: Probably with rapid recovery.

- Line 209: “At 3 months, 484/504 patients were evaluated (Figure 2).”Comments, proposals:  Probably 484/569 (please add %); reference to Figure 2 would better fit at the end of paragraph.

- Line 226, Table 4: Delay 1st symptoms - 1st consultation at the TBD-RC. Comments, proposals: Why did the authors choose these time-frames? The time ranges overlap (for example: 0-155 days, 155-512 days) - please adjust.

- Line 228: “3.3. Factors associated to recovery at a later point in time” Comment: Probably with recovery.

- Lines 23 –232: The meaning of the sentence “Only patients with a complete recovery at 3 and/or 6 months were considered in recovery at 9-12 months in the absence of clinical evaluation” is not clear.

- Line 234: “…among the 4 diagnostic groups (p=0.001)” Proposal: omit diagnostic.

- Lines 241–243: Please mention also other statistically significant parameters.

- Lines 250 –257: The first part of the paragraph should be revised.

Discussion

- Lines 259 –270:  4.1. Summary of the principal findings. Proposal: Add also discussion on findings at 9-12 months follow-up.

- Lines 282–286: A possible (additional) interpretation of the findings would be that patients with confirmed LB have more chances to be cured but for some presentations it may take >3 months till improvement.

- Lines 287–290: “Few patients (9%) had no specific diagnoses compared to other studies (27–30) probably because they had a regular follow-up in the adapted care sector since the beginning of their management, which allowed for clinical re-evaluations and the confirmation of the diagnosis for a longer period.” Please give proportions of patients without specific diagnosis in the other studies.

- Lines 296 –297: “The predominance of functional symptoms had been demonstrated in previous studies, as had facial palsy in confirmed LB (27–30), but no studies ….” Comment: Please revise this sentence.

- Line 302: Among them instead of among whom.

- Lines 310-312: What is the meaning of  the sentence: “The proportion of patients with non-recommended antibiotic therapy was significantly higher in the group PTLDS/sequelae, suggesting an initial misdiagnosis or causative role.”

- Lines 322–323: Better clinical outcome instead of better clinical evolution of the patients.

- Line 338: “… between 15 March and 30 April“ - year?

Author Response

Yours sincerely

The authors

Reviewer 2 Report

Hi,

I began going over this manuscript line by line, but there are too many flaws in the basic study design to continue any further.

Line 62: what other pathologies?

64: I don’t see how these two references support evidence of functional symptoms 6,7

65: “well conducted treatment” is an inaccurate statement 5-8

66 Rare is incorrect

67 Rare is inaccurate

68-71: inaccurate

73: incorrect

100: Non-recommended antibiotic therapy was defined as longer than 8

weeks and/or associated antibiotics (≥2 molecules prescribed concomitantly). This created an improper selection bias towards the more severe cases to be placed into this group.

105: what symptoms were assessed as a part of the clinical exam?

114: lack of clarity differentiating PTLDS vs. sequelae groups

124: how were they evaluated?

162: How did the authors know the conditions labeled as differential diagnosis were not caused by Bb infection?

Chart: Optic nevitis should be optic neuritis, were rickettsiosis, tularemia the only tickborne coinfections tested for? What about the more common tick-borne coinfections?

What in the world is Anxio-depressive syndrome?

Bodily Distress Syndrome is a highly controversial and invalid as a diagnosis.

Many of the conditions on this chart are conditions well recognized to be caused by Bb infections in the medical literature.

Author Response

Yours sincerely

The authors

Reviewer 3 Report

Good and clear multidisciplinary approach for LB differential diagnostics.  Recommend for the future to investigate inflammatory process of LB pathogenesis from the point of immunology. 

Author Response

Yours sincerely

The authors

Reviewer 4 Report

Review of the study titled ”Multidisciplinary management of suspected Lyme borreliosis: 2 clinical features of 569 patients, and factors associated with re-3 covery at 3 and 12 months, a prospective cohort study”
The study describes the flow of patient in a clinical center called “Tick-Borne Diseases Reference Center” during 3 years from 2017 to 2020. Even if true tick transmitted diseases are quite rare and in most cases mild transient disease, there are risk perceptions in society displayed on social media about tick bites frequently leading to serious health problems. This has also attracted political attention.  As a consequence  “specialized” clinical centers have been established to take care of patients suspected of a tick borne disease  (TBD), especially Lyme borreliosis, also in other countries. The experience in this paper, in line with other similar published clinical series, is that the majority of cases do not have a tick borne disease. And a multidisciplinary approach is thus important. 
The importance in naming such a “TBD –centre”, even if most diagnoses belong to other specialities, lies in establishing an alternative to certain alternative clinics exploiting societal concerns of TBDs. These clinics may not perform proper multidisciplinary clinical approach, and as proper diagnosis is delayed, this may cause harm to patients. 
Another good reason to describe such a broad case series, is that most clinical publications are more narrow within specialities, thus descriptions of multidisciplinary clinical care is important. This helps guide clinical work of which diagnoses to consider in patients where there are no clear pointers, at the outset, for a clear referral to specialized care.  
A very thorough clinical evaluation was performed at the centre, and is described in the paper, with follow up.  This reviewer is not aware of another similar study describing follow up as well with final diagnosis.
Thus, there are good general and specific reasons to publish the present study.
The main findings are that the majority of cases do not have LB or another TBD. That most cases improve, regardless of final diagnosis. That the multidisciplinary approach was efficient in establishing final diagnoses with a few weeks. 
Strengths
Large number of cases with quite complete follow up. Thus final diagnoses are included

Weaknesses
Not really, beyond the more general that summaries of recovery are somewhat problematic concerning a mixture of diagnoses. 

Detailed comments
Page 2
Line 69: PCR is mentioned, but not in the rest of the paper?  Was PCR used at all?  Suggest to add a comment about PCR. It is not clinical routine in the reviewers hospital.  Was it usefull?
Page 3: 
Linew 102. “Molecules”  What is meant ? different drugs ? Types of antibiotics ?
Line 121-122: The sentence with “adapted sector”. What is meant by the term?  Relevant medical specialities ?  Infectious diseases, neurology etc.  The same as “final orientation” in figure 1. The term is used several times in the manuscript.
Lines 127-131:  It is appreciated by the reviewer that a simple and direct recovery classification is used (instead of some general purpose “standardized” questionnaire). Perhaps explain a bit more if this was a questionnaire to the patient, or a kind of joint conclusion between clinician and the patient. 
Section on patient data: Perhaps add a further, but short explanation about the “standardized medical files”.  In the opinion of the reviewer it is a strength of the study that data is collected in a systematic manner as part of the clinical routine. The data in the tables 2+3 are very detailed. Was all info collected on all cases, or rather only relevant details on each individual especially concerning table 3. Where all 569 cases interviewed about all variables polymyalgia, polyarthralgia etc.?
Line 136 ff section on statistical analysis.  In the tables 4 and 5 there are some binning of data. Agegroups  <35, 35-48, etc.  Delays in days 0-155…..<1393, 0,1-15 etc. Why and how were these intervals chosen ? To make groups of nearly equal size for the regression model? Suggest to explain this. 
Page 4
Lines 166: “diagnoses associated with confirmed LB”  What is meant here?  Concurrent/coexisting?
The reviewer is not aware of comorbidities associated with LB as with other diseases, for example diabetic complications?  Some patients may have sequelae. Which of the diagnoses in table 1. Bis are considered “associated” ?  Suggest to explain and rephrase. 
Line 169  “a mean of 1.7 diagnoses”  Such a use of central tendency makes little sense. Is this data from table 4 at the follow up or for all 569 cases. Suggest to state more directly the data xxx patients had a single diagnosis xxx had two or more diagnoses. 
Line 182. Table 1.  Consider phrasing “Failure of the antibiotics testing”.  Maybe “trial” instead of “testing”?  Or  Antibiotic treatment, no clinical response.  
Lines 184-6: The reviewer does not really like this approach, even if frequently used in clinical practice. A response cannot confirm a diagnosis, a missing response does not rule out, as these events may also occur spontaneously with or without infection with B. burgdorferi. 
Suggest to delete “ruled out….and” Then the sentence could be: “Initial suspicion of probable LB but patients did not improve after the antibiotic therapy, which led us to consider another diagnosis.”
Page 6  Table 2.
“Letter from a physician to address the patient” Consider English wording, what is meant here? 
Suggest “Patient referred from”

“No letter, patient consulting by himself” perhaps change to “Patient self-referral”.
Clinical signs/symptoms implicating early disseminated LB (> 6 months)   “<”
Maybe “indicating” instead of “implicating”
Page 7 Table 3. 

Page 8:  Figure 1.  Evolution ? You mean “evaluation” ? in the lower boxes.

Page 13 lines 262 to 268  
This is an important paragraph, which could be mis-interpreted by some readers. It is important to show that a primary diagnosis of definite LB/possible LB may be found inappropriate – on second evaluation.
Suggest to be very clear here if these diagnoses are really “linked” to LB – and not overlooked because of too much primary focus on TBDs. The reviewer is critical about the construction of this type of TBD centres for political reasons, rather than diagnostic centers handling difficult to diagnose cases in a more general manner – focusing on NOT overlooking malignant disease or other serious disease. May the focus on rather rare “tick borne diseases” delay the more final appropriate diagnosis? Was the lymphoma overlooked at baseline?  Did the 2 persons with “previous neuro-degenerative” receive treatment for LB. In the experience in my own clinical setting the Neurologists insist on diagnostics for Lyme neuroborreliosis, even when clinically unlikely, before referral from infectious diseases. This may cause delay. And, may save some clinical work as many cases improve spontaneously.
Bottomline:
Title: Description of patients with stagnation or deterioration at 3 months follow up with a primary diagnosis of LB. 
Suggest: In the text make more clear these alternative diagnoses are an important finding and the good reason to have a multidisciplinary clinical management.

Page 14 lines 302-3.  What is adapted care sector ? Another clinic, primary care?
Lines 337 to 343. + line 351. See also comment above about Page 13 lines 262 to 268
This is an important discussion of the relevance of establishing a TBD-RC. This reviewer has some difficulty with appreciating this type of clinic. First of all at least 70% of cases have no relevance to a tick-borne ailment. Second of all it is an endless task open up new clinics according a changing public focus, lobbying patient organisations, rather than medical need: chronic fatique clinics, Post-covidclinics  occult cancer etc. 
In my opinion rather more general multidisciplinary clinics are needed, and may more easily be established at larger number of regional centres throughout the country? 
Nearly any disease may present with a subtle, slow onset uncharacteristic clinical course?       
Line 351. Something like this is discussed already line 342-3.  But suggest to be more precise what you mean. Clinics with a more general scope, or perhaps screening session before the larger multidisciplinary effort ?

Round 2

Reviewer 1 Report

The authors adequately responded to all critiques.

Author Response

Thank you very much for your very good and useful reviewing.

The authors

Reviewer 2 Report

Line 2: Multidisciplinary management of suspected Lyme borrelioseis: 3 clinical features of 569 patients, and factors associated with recovery at 3 and 12 months, a prospective cohort study

Change to:

Multidisciplinary management of suspected Lyme borreliosis: 3 clinical features of 569 patients, and factors associated with recovery at 3 and 12 months, a prospective cohort study based solely upon the surveillance laboratory criteria

Line 64: Delete: . Some functional symptoms 65 may be present at all stages, which can further complicate the diagnosis (6,7). Invalid statement

Line 66: delete: well-conducted

Line 67: delete Rare

Line 68: delete Rare

Line 72: delete: Current diagnostic tools are performant if their indication and interpretation are well-respected; otherwise, they may lead to the wrong diagnosis

Line 74-77 delete, unsupported

The authors relied excessively upon laboratory criteria for diagnosis and failed to adequately consider multisystem symptoms caused by Lyme borreliosis. The two-tiered surveillance criteria has never been validated as an adequate marker that could rule out the presence of active or latent infection. Lyme borreliosis symptoms were incorrectly labeled as functionals symptoms. This article advances a false narrative that will harm patients with Lyme disease. I recommend rejection of this article due to flawed assumptions in the study design.

Author Response

Dear reviewer,

Thank you very much for the time spent to review this manuscript.

As mentioned in our previous answers, we do not agree with your suggestions that are not supported by the literature, and we cannot modify our manuscript.

Moreover, the outcomes of the patients are based on clinical evaluations only and not on laboratory findings. Only 26% of the patients had a positive serological test, which demonstrates that we did not consider only the laboratory findings: clinical manifestations at baseline and clinical outcomes were the most important.

Yours sincerely,

The authors